# Poly(ethylene glycol) Diacrylate Iongel Membranes Reinforced with Nanoclays for CO_2_ Separation

**DOI:** 10.3390/membranes11120998

**Published:** 2021-12-20

**Authors:** Ana R. Nabais, Rute O. Francisco, Vítor D. Alves, Luísa A. Neves, Liliana C. Tomé

**Affiliations:** 1LAQV-REQUIMTE, Department of Chemistry, NOVA School of Science and Technology, FCT NOVA, Universidade Nova de Lisboa, 2829-516 Caparica, Portugal; a.nabais@campus.fct.unl.pt (A.R.N.); rr.francisco@campus.fct.unl.pt (R.O.F.); 2LEAF—Linking Landscape, Environment, Agriculture and Food—Research Center, Associated Laboratory TERRA, Instituto Superior de Agronomia, Universidade de Lisboa, Tapada da Ajuda, 1349-017 Lisabon, Portugal; vitoralves@isa.ulisboa.pt

**Keywords:** ionic liquids, UV cross-linked polymer network, nanoclays, iongels, hybrid organic-inorganic membranes, CO_2_ separation

## Abstract

Despite the fact that iongels are very attractive materials for gas separation membranes, they often show mechanical stability issues mainly due to the high ionic liquid (IL) content (≥60 wt%) needed to achieve high gas separation performances. This work investigates a strategy to improve the mechanical properties of iongel membranes, which consists in the incorporation of montmorillonite (MMT) nanoclay, from 0.2 to 7.5 wt%, into a cross-linked poly(ethylene glycol) diacrylate (PEGDA) network containing 60 wt% of the IL 1-ethyl-3-methylimidazolium bis(trifluoromethylsulfonyl)imide ([C_2_mim][TFSI]). The iongels were prepared by a simple one-pot method using ultraviolet (UV) initiated polymerization of poly(ethylene glycol) diacrylate (PEGDA) and characterized by several techniques to assess their physico-chemical properties. The thermal stability of the iongels was influenced by the addition of higher MMT contents (>5 wt%). It was possible to improve both puncture strength and elongation at break with MMT contents up to 1 wt%. Furthermore, the highest ideal gas selectivities were achieved for iongels containing 0.5 wt% MMT, while the highest CO_2_ permeability was observed at 7.5 wt% MMT content, due to an increase in diffusivity. Remarkably, this strategy allowed for the preparation and gas permeation of self-standing iongel containing 80 wt% IL, which had not been possible up until now.

## 1. Introduction

In the last 25 years, numerous studies have been focusing on ionic liquids (ILs), a material with exceptional features that keeps stimulating researchers’ curiosity today. Besides being liquid salts at room temperature due to their melting points below 100 °C, ILs are characterized by distinct tunable properties [1,2]. The most appealing property of ILs is their designable nature, which allows the preparation of task-specific materials for the envisioned application by the combination of different ions and the addition of specific functional groups [1]. Regarding energy-related applications, their ionic conductivity, low volatility, electrochemical and thermal stability serve as advantages in the development of batteries [3], fuel cells [4], and chemical sensors [5]. On the other hand, considering the unique properties of some ILs, such as biocompatibility, biodegradability, and bioactivity, biomedical applications such as biosensors [6] and drug delivery systems [7] have also been developed. This work focuses on the use of ILs for carbon dioxide (CO_2_) separation from light gases taking advantage of the interactions established between the electrical charges of the IL and the quadrupole of CO_2_ molecules, and combining them with membrane technology, an economic and environmentally friendly approach to CO_2_ capture [8,9,10].

In the last years, diverse ionic liquid-based membranes have been developed. In particular, iongel membranes have shown promising results for CO_2_ separation, with performances similar to those of supported ionic liquid membranes (SILMs), surpassing the 2008 Robeson upper bound [8]. However, SILMs often have stability issues brought up by operation conditions (e.g., high pressures and high temperatures) that can cause the IL displacement from the porous support [8,9]. Instead, iongels are soft solid materials that present a gel-like nature, as they are mainly composed of ILs, retaining a liquid-like gas transport [11]. For the fabrication of iongels, there is a wide variety of materials that can be selected, as well as diverse routes to approach their preparation [11]. In order to ensure that high IL contents (>50 wt%) are efficiently retained within the iongel structure, different types of gelators can be used [11], such as polymer networks [12], block copolymers [13], biopolymers [14], supramolecular gelators [15], organosilane networks [16], or inorganic nanoparticles [17]. The combination of ILs with polymers, which are commercially available such as poly(vinylidene fluoride) (PVDF) [18] or poly(ethylene glycol) (PEG) [19], and poly(ionic liquid)s (PILs) with imidazolium [20] or pyrrolidinium [21] cations and a variety of counter anions, is perhaps the most straightforward approach to fabricate iongel membranes. PIL-based membranes can hold up to 60 wt% IL due to the advantage of establishing strong electrostatic interactions between their charged PIL and IL components [8]. In order to further improve their CO_2_ separation performance along with other properties, some researchers also reported the incorporation of porous fillers, such as zeolites [22] and metal organic frameworks (MOFs) [23] to produce mixed matrix membranes (MMMs). Additionally, significant efforts have been made in the fabrication of iongels using cross-linked PIL-based networks since this strategy provides extra mechanical stability and allows the incorporation of higher IL contents [24]. Nevertheless, the preparation routes of cross-linked PIL-based iongels are often complex, involving multi-pot polymerization procedures or several steps of synthesis and purification at the monomer level.

A simpler strategy is the preparation of iongels by single-pot polymerization using commercially available non-ionic polymer networks, such as poly(ethylene glycol) (PEG)-based networks, due to their high affinity towards CO_2_ [25]. The polymerization can be thermal or UV initiated. Recently, Martins et al. [12] studied the influence of the anion structure on the CO_2_/N_2_ separation performance of UV-cross-linked poly(ethylene glycol) diacrylate (PEGDA) iongels. Using imidazolium-based ILs with cyano-functionalized ([C(CN)_3_]^–^ and [B(CN)_4_]^–^) or fluorinated anions ([TFSI]^–^ and [FSI]^–^), self-standing iongels containing between 60 and 90 wt% of IL were obtained. Although the PEGDA iongel with 70 wt% [C_2_mim][C(CN)_3_] IL showed CO_2_ permeability of 583 barrer and ideal CO_2_/N_2_ selectivity of 66, further improvements are needed to circumvent the use of a porous support and preserve the iongel mechanical stability while increasing the IL content.

One popular strategy used to improve the mechanical stability of polymer membranes is the incorporation of fillers [26]. A wide variety of fillers, porous or non-porous, have been reported in the literature [27,28], including the use of nanoclays [29]. Nanoclays are natural, nonporous inorganic materials that have been considered to enhance the mechanical properties of polymeric membranes, as well as their gas separation performances [30,31]. Several studies revealed that the addition of low contents of nanoclay particles (≤10 wt%) can improve the thermal, mechanical, electrical, and barrier properties of MMMs [32], because of the attractive properties of nanoclays such as ion exchange capacity and specific surface area, besides being widely available at low cost [33]. Montmorillonite (MMT) is one of the most commonly used nanoclays, mainly due to its ability to be dispersed into polymeric matrices. This nanoclay has a 2:1 phyllosilicate structure, composed of 1 nm layers of alumina octahedra sheets placed between two sheets of silica tetrahedra [34,35].

This work focuses on the development of robust PEGDA iongel membranes reinforced with MMT nanoclay particles, considering a simple one-pot preparation method. The [C_2_mim][TFSI] IL was selected and firstly used at 60 wt% IL content, seeing that previously reported PEGDA iongels have shown higher storage modulus when comprising this IL rather than other ILs having different anions (i.e., [FSI]^–^, [C(CN)_3_]^–^ and [B(CN)_4_]^–^) [12]. Moreover, low contents of MMT nanoclay particles were used, between 0.2 and 7.5 wt%, in order to fabricate self-standing iongels with good mechanical properties. All the prepared iongel materials were characterized by several techniques to study their physico-chemical properties. Pure gas permeation experiments were also carried out to evaluate the potential of PEGDA iongels reinforced with nanoclays as CO_2_ separation membranes.

## 2. Experimental Section

### 2.1. Materials

The ionic liquid 1-ethyl-3-methylimidazolium bis(trifluoromethylsulfonyl)imide ([C_2_mim][TFSI], 99 wt% pure) was acquired from IoLiTec GmbH (Heilbronn, Germany). Sigma-Aldrich (Madrid, Spain) supplied the poly(ethylene glycol) diacrylate (PEGDA, M_n_ 575 g mol^−1^) and 2-hydroxy-2-methylpropiophenone (DAROCUR, 97 wt% pure). The nanoclay Nanomer^®^ I.34TCN, montmorillonite (MMT) clay surface modified with 25–30 wt% methyl dihydroxyethyl hydrogenated tallow ammonium was obtained from Sigma-Aldrich (Taufkirchen, Germany). The gases carbon dioxide (CO_2_, 99.998 wt% pure), nitrogen (N_2_, 99.99 wt% pure), and methane (CH_4_, >99.99 wt% pure) were provided by Praxair (Almada, Portugal). Hydrogen (H_2_, 99.99 wt% pure) gas was obtained from Air Liquide (Almada, Portugal).

### 2.2. Preparation of the Iongels

The neat iongel and the iongels reinforced with different MMT contents (0.2, 0.5, 1.0, 2.5, 5.0, and 7.5 wt%) were prepared by UV-initiated free radical polymerization (Figure 1). First, the desired amounts of MMT were added to 60 wt% of [C_2_mim][TFSI] IL. The solutions were left stirring overnight at room temperature on a magnetic stirrer plate (Magnetic Emotion Mix 15 eco, from 2mag, Munich, Germany) and sonicated for 1 h in an ultrasonic bath (Sonorex Digitec, from Bandelin, Berlin, Germany). Afterwards, the necessary content of PEGDA was added into the solutions and magnetically mixed for 1 h at room temperature. Next, the radical photoinitiator DAROCUR (3 wt% to PEGDA) was added, and the solutions were mixed and poured between two Rain-X™ coated quartz plates. The plates were separated by a spacer to assure the desired thickness and diameter of the iongel membrane and secured in place using clips. The solutions were then exposed to UV radiation with a wavelength of 365 nm and an intensity of 1.7 mW cm^−2^ for 10 min. Lastly, the iongels were carefully peeled from the plates and stored in Petri dishes until characterization. The iongel samples are systematically designated as “60 TFSI-XX PEGDA-YY MMT”, according to PEGDA (XX) and MMT (YY) content.

### 2.3. Characterization Methods

Scanning Electron Microscopy (SEM) was carried out in order to examine the morphology of the iongel membranes. The samples were covered with a thin layer of Au-Pd before the acquisition of the images by a JOEL scanning electron microscope, model 7001F (USA) with an electron beam intensity of 10 kV.

Attenuated Total Reflectance Fourier-Transform Infrared (ATR-FTIR) spectroscopy was performed to confirm the incorporation of the IL, PEGDA, and MMT into the iongel membranes, verify possible interactions established between them, and confirm the occurrence of the photopolymerization reaction. The spectra were obtained using a Perkin Elmer Spectrum Two spectrometer that collected 10 scans from 400 to 4000 cm^−1^.

The contact angles were measured to assess the hydrophilicity of the iongel membranes, using a sessile drop method. A drop of distilled water was manually deposited on the surface of each iongel. For each sample, 10 frames were acquired with an intercalation of 1 s and processed by CAM100 (KSV Instruments LTD, Helsinki, Finland) software. The mean contact angle was obtained when the drop shape was fitted to mathematical equations.

Thermogravimetric Analysis (TGA) addressed the thermal stability of samples (ca. 10 mg) of the starting materials and the iongel membranes by determining their onset temperatures (*T_onset_*). The measurements were performed using a Lass Evo TGA-DTA/DSC 1600 °C PG from Setaram KEP Technologies (Caluire, France) from 25 to 600 °C, at a heating rate of 10 °C min^−1^, under an argon atmosphere.

The mechanical properties of the iongel membranes were determined by puncture tests. The tests were performed using a TA XT Plus Texture Analyzer (Stable Micro Systems, Godalming, UK) by puncturing each membrane through a hole (diameter of 10 mm) with a cylindrical probe with 2 mm of diameter at a constant velocity of 1 mm s^−1^, at room temperature. The data obtained allowed the determination of the puncture strength according to Equation (1)
(1)$$\sigma =\frac{F}{A}$$
where *σ* is the puncture strength (Pa), *F* is the maximum force applied by the probe (N) and *A* is the cross-sectional area of the probe (m^2^). In order to compare the results rigorously, the puncture strength was normalized with thickness following Equation (2)
(2)$$\sigma_{n}=\frac{\sigma }{l}$$
where $\sigma_{n}$ is the normalized puncture strength (MPa/mm) and *l* is the thickness of the iongel (mm). The elongation at break given by Equation (3) was also determined
(3)$$\varepsilon \;\left(\%\right)=\left(\frac{\sqrt{h^{2}-d^{2}}-h}{h}\right)\times 100$$
where ε is the elongation at break (%), *h* is the radius of the iongel exposed in the cylindrical hole of the sample holder (mm) and *d* is the distance of the probe from the point of contact to the point of puncture (mm). The Young’s modulus of all prepared iongels was obtained from the initial slope of the graphic representation of the puncture strength (*σ*) as a function of the elongation at break ($\varepsilon \;\left(-\right))$. For each sample, at least four replicates were performed and the mean values of puncture strength, elongation at break, and Young’s modulus were calculated.

### 2.4. Gas Permeation Experiments

Pure gas permeation experiments using CO_2_, N_2_, CH_4,_ and H_2_ were performed on a gas permeation setup described elsewhere [36]. The setup is composed of a cell immersed in a water bath maintained at 30 °C by a thermostat (Corio C, from Julabo, Seelbach Germany). The pressure of each cell compartment is measured over time by a transducer (Jumo type 404327, Germany) and recorded by the software PicoLog. The compartments were pressurized with pure gas and, when the pressure was stable, the experiments began with the application of a transmembrane driving force of around 0.7 bar of relative pressure. The data collected allowed the determination of the permeability of the iongels according to Equation (4)
(4)$$\frac{1}{\beta }\times \ln\left(\frac{\Delta p_{0}}{\Delta p}\right)=P\times \frac{t}{l}$$
where Δ*p_0_* and Δ*p* (bar) are the pressure variations between the feed and the permeate compartments of the cell at the beginning of the experiment and over time, respectively, *P* is the permeability of the iongel (m^2^ s^−1^, where 1 barrer = 1 × 10^−10^ cm^3^ (STP).cm.cm^−2^.s^−1^.cmHg^−1^ = 8.3 × 10^−13^ m^2^ s^−1^), *t* is time (s), *l* is the thickness of the iongel (m) and *β* is the geometric parameter of the cell (m^−1^), given by Equation (5)
(5)$$\beta =A\times \left(\frac{1}{V_{feed}}+\frac{1}{V_{perm}}\right)$$
where *A* is the membrane area (m^2^), *V_feed_* and *V_perm_* are the volumes of the feed and permeate compartments (m^3^), respectively. The permeability of the iongel was obtained from the slope when *1/β* ln*(*Δ*p_0_/*Δ*p)* was plotted as a function of *t/l*. The ideal selectivity of the iongels is given by Equation (6)
(6)$$\alpha =\frac{P_{A}}{P_{B}}$$
where *α* is the ideal selectivity, *P_A_* and *P_B_* are the permeabilities of the most permeable gas and the least permeable gas, respectively.

## 3. Results

### 3.1. Scanning Electron Microscopy (SEM)

SEM images of the cross-section of the iongels containing 0, 0.2, and 7.5 wt% MMT are displayed in Figure 2a,b,c, respectively. The images obtained for the remaining iongels are represented in Appendix A–d of Appendix A. Dense morphologies were obtained in all iongels, regardless of the MMT content. As the MMT content increases, the small inorganic particles become more visible, and at 7.5 wt% MMT, some agglomerates are clearly distinguishable. This morphology is a consequence of, not only the high MMT concentration, but also the absence of a solvent during iongel preparation, which can difficult the dispersion of MMT particles. Despite the agglomerates, no major defects that could compromise the performance of the iongels reinforced with MMT were observed. It should also be noted that, apart from the neat iongel image, some reliefs can be seen in the iongels structure, due to the fracturing process, under liquid nitrogen.

### 3.2. Attenuated Total Reflectance-Fourier Transform Infrared Spectroscopy (ATR-FTIR)

ATR-FTIR spectroscopy was used to confirm the structure of the iongels components, as well as to detect any possible interactions being established between them. The obtained FTIR spectra of the 60 TFSI-40 PEGDA iongel, MMT powder, and 60 TFSI-32.5 PEGDA-7.5 MMT iongel are displayed in Figure 3. In the 60 TFSI-40 PEGDA spectrum, the peaks detected at 1053, 1133 and 1179 cm^−1^ correspond to the [TFSI]*^–^* anion, while the bands at 3120 and 3160 cm^−1^ correspond to the CH_3_ bending and CH_2_ stretching vibrations of the [C*_2_*mim]*^+^* cation, respectively. The peak at 1732 cm^−1^ is attributed to the C=O symmetric stretching, while the bands between 2866 and 2936 cm^−1^ are attributed to the C-H stretching of PEGDA. The IR spectra of the pristine [C_2_mim][TFSI] IL and PEGDA can be found in Appendix A of Appendix A. The absence of visible peaks at wavenumbers of 1619 and 1635 cm^−1^, arising from the terminal acrylate groups of PEGDA indicates the high extension of the polymerization reaction. The same peaks and bands were also observed on the 60 TFSI-32.5 PEGDA-7.5 MMT iongel spectrum. Regarding the MMT spectrum, peaks at 517 and 1012 cm^−1^ are representative of the Si-O bending and stretching of the silica tetrahedra sheets, respectively. At 779, 885, and 918 cm^−1^ the peaks observed are attributed to the bending of AlMgOH, AlFeOH, and AlAlOH functional groups, respectively. The observed peaks at 2851 and 2924 cm^−1^ are attributed to the stretching of the C-H bending of the CH_2_ groups and bending of CH_3_ functional groups, respectively, while at 3630 cm^−1^ a peak from the stretching of OH groups is associated with Al-OH and Si-OH from alumina and silica sheets. Due to overlapping, the characteristic peaks of MMT are not easily distinguishable in the spectrum of the reinforced iongel. Nonetheless, the influence of MMT in the 60 TFSI-32.5 PEGDA-7.5 MMT iongel spectrum can be observed at 450, 1053, and 2924 cm^−1^.

### 3.3. Contact Angle Measurements

The water contact angles of iongels reinforced with different MMT contents were determined and the results are depicted in Figure 4. A contact angle of 41.3° was obtained for the 60 TFSI-40 PEGDA iongel membrane. As it can be seen, at lower contents (≤1 wt%), the incorporation of the MMT particles did not have a significant impact on this parameter. On the other hand, at higher contents, there is a clear increase in the contact angle, from 46.9° for the 60 TFSI-37.5 PEGDA-2.5 MMT iongel, up to 58° for the highest MMT content (7.5 wt%). These variations at higher MMT contents may be attributed to two reasons; first, a tendency for the MMT to lower the hydrophilicity of the iongels, and/or the decrease in the PEGDA content. PEGDA is composed of hydrophilic ethylene oxide units and hydrophobic acrylate units. Longer PEGDA networks, such as the one used in this work, present higher contents in ethylene oxide groups, making the material more hydrophilic. A decrease in the hydrophilic PEGDA content, while maintaining the (hydrophobic) IL composition, results in a decrease in the hydrophilic character of the iongels reinforced with MMT. Nonetheless, all the prepared iongels presented a hydrophilic nature, regardless of the MMT content. This comes as an advantage for gas transport since the solubility of CO_2_ in hydrophilic polymers is reported as being higher than in hydrophobic ones [37].

### 3.4. Thermogravimetric Analysis (TGA)

The *T_onset_*, considered as the temperature at which the baseline slope changes, of MMT, [C_2_mim][TFSI] IL, PEGDA and all prepared iongels with different MMT contents are listed in Table 1. The respective thermogravimetric profiles of all iongels prepared in this work, as well as of the neat components can be found in Appendix A of Appendix A, respectively. The *T_onset_* obtained for [C_2_mim][TFSI] IL, PEGDA and MMT are in accordance with previous reports [12,38,39,40].

The degradation of MMT started around 200 °C and from this point until around 530 °C this degradation step is associated to the release and decomposition of the MMT organic modifier, methyl dihydroxyethyl hydrogenated tallow ammonium. The two decomposition steps observed in this temperature range are related to the different sites where the functionalization of the nanoclay may occur: at the external surface and at the edges (first step) and in the interlayer space (second step) [38].

As it can be seen from Table 1, until 2.5 wt% of MMT content there is no significant change in the *T_onset_* of the prepared iongels reinforced with nanoclays, due to its low content. However, at temperatures up to 200 °C, a more pronounced decomposition of around 5% was noticed in the iongels containing different contents of MMT, compared to the neat iongel (Appendix A of Appendix A). This happens probably due to the lower thermal stability of the MMT nanoclay, which started to decompose at around 200 °C. For all the iongels, a second decomposition step can be observed at around 400–410 °C, which is associated to the degradation of both PEGDA and IL. The *T_onset_* decreased from around 339 °C for the iongels with up to 2.5 wt% MMT to 331 and 326 °C for the iongels with 5 and 7.5 wt% MMT content, respectively, due to the higher content of MMT with a lower thermal stability. In sum, it can be concluded that the thermal stability of the iongels is influenced by the incorporation of higher contents of MMT. Furthermore, the chosen nanoclay can be considered appropriate to be incorporated into iongels since its weight remained constant up to 150 °C.

### 3.5. Mechanical Properties

The normalized puncture strength, elongation at break, and Young’s modulus of the iongels with different MMT contents can be found in Figure 5a–c, respectively. It is known that the incorporation of nanoparticles, such as nanoclays, into a polymeric matrix, can improve the mechanical stability of the resulting materials. As such, it could be expected an increase of puncture strength and deformation of the iongels prepared in this work, if the MMT particles are well dispersed within the polymer matrix.

From Figure 5a, it is clear that the puncture strength increases in iongels with low MMT contents (between 0.2 and 1 wt%), where a maximum increase of 49% was achieved for 1 wt% MMT (6.0 MPa/mm). This suggests that the MMT particles are evenly dispersed throughout the iongel, acting as a reinforcement, while the material becomes more resistant to the external force applied during the puncture tests.

On the other hand, MMT contents higher than 1 wt% resulted in a reduction in the puncture strength to 2.98 MPa/mm for a content of 7.5 wt% MMT, even lower than the neat iongel. It seems that higher MMT contents can result in a lower level of dispersion of the particles and a higher number of agglomerates, as it can be seen in Figure 2. The appearance of weak spots due to the aggregation of MMT sheets consequently result in lower mechanical resistance of the material. In previous works, where MMT particles were incorporated into polymeric matrices, such as polyethersulfone and cellulose, higher MMT contents (>5 wt%) also resulted in deprived mechanical performances [41,42].

A similar trend was obtained regarding the elongation at break of the prepared iongels, as presented in Figure 5b. The elongation at break increased 57%, from 10.7% deformation for the neat iongel to 16.8% for the iongel reinforced with 1 wt% MMT. At lower MMT contents, and consequently homogeneous dispersion of the particles, there is a possible rearrangement of the clay sheets in the direction of the deformation, allowing higher elongation [41]. Similar to what was observed for the puncture strength, higher MMT contents decrease the ability of the iongels to elongate before rupture, due to the aforementioned lower degree of dispersion of MMT particles.

Figure 5c represents the variation in Young’s modulus with the MMT content incorporated into the iongels. Young’s modulus, defined as the ratio between the puncture strength and deformation is, in its essence, the resistance of the material to deformation when a force is applied. A significant increase in Young’s modulus was achieved for all the iongels reinforced with MMT, compared to the neat iongel, with a maximum at 5.1 MPa for 1 wt% content. This improvement is attributed to the good dispersion and alignment of MMT sheets within the iongel and the additional rigidity provided by the inorganic particles. The slight decrease in Young’s modulus at higher contents (>1 wt%) is again attributed to a reduced level of dispersion of the MMT particles. Nonetheless, the iongels reinforced with MMT particles revealed better overall mechanical performances, compared to the 60 TFSI-40 PEGDA iongel, with stronger membranes upon puncture achieved for 1 wt% MMT content.

### 3.6. Gas Permeation Experiments

The CO_2_ permeability, as well as the ideal CO_2_/N_2_, CO_2_/CH_4,_ and CO_2_/H_2_ selectivities are plotted in Figure 6 and Figure 7, respectively. At 0.2 wt% MMT content, there are no significant variations in the CO_2_ permeability, probably due to such lower MMT content. The further increase in the MMT content resulted in a continuous increase in gas permeability. The CO_2_ permeability increases from 40 barrer for the 60 TFSI-40 PEGDA iongel up to 98 barrer at 7.5 wt% MMT content. Following the same trend, the N_2_, CH_4,_ and H_2_ also increase with higher MMT concentrations (Appendix A).

In order to study the effect of the MMT particles in the gas transport of the prepared iongel membranes, the CO_2_ solubility of the MMT powder was also measured, following a pressure decay method, described elsewhere [36]. It was not possible to observe any pressure decay over time (40 h), so it can be assumed that the MMT particles do not show affinity towards CO_2_. Therefore, the incorporation of MMT particles likely provided an increase in diffusivity, without any substantial influence in terms of solubility. Nonetheless, the CO_2_ permeability is considerably higher than that of the remaining gases. The CO_2_ is a highly condensable gas and has a high affinity for the iongels, due to the fact that the quadrupole moment of the CO_2_ molecules interacts favorably with the electrical charges of [C_2_mim][TFSI] IL, which is the major component of the iongels [8]. Apart from the solubility, the smaller kinetic diameter and linear molecular shape of CO_2_ are also advantageous for fast diffusion of the gas across the iongel [31]. It can be concluded that, while the CO_2_ permeability is governed by a combination of diffusivity and solubility, the permeation of the remaining light gases in this case is mainly controlled by diffusivity.

Looking at Figure 7, it can be seen that the ideal selectivities of all gas pairs are improved by the incorporation of up to 0.5 wt% MMT. In this work, the use of a modified MMT clay is expected to improve the compatibility between this inorganic filler and the polymeric phase, allowing a more even distribution of the particles across the iongel matrix [43]. At lower contents and good MMT particles distribution, the selectivity can be enhanced by a shape and size discrimination of gases passing between the intercalated clay layers.[40] On the other hand, at higher contents, the iongels reinforced with MMT suffer a decline in selectivity. The formation of agglomerates (Figure 2) increases the free volume of the reinforced iongels, thus increasing gas diffusion. This will allow all gases to be transported more easily across the iongel membranes, resulting in the observed decrease in CO_2_/N_2_, CO_2_/CH_4,_ and CO_2_/H_2_ selectivities [31]. Similar results have also been reported in the literature. For instance, Jamil et al. [44] found an optimal modified-MMT content at 2 wt% to prepare polyetherimide-based mixed matrix hollow fiber membranes. Hashemifard and co-workers [40] found out that an increase in selectivity and CO_2_ permeance could be achieved by incorporating up to 1 wt% of Cloisite 15A. However, at higher contents, a dramatic decline in the separation performance was observed, due to an increase in clay agglomeration.

To study the possibility of further increasing the IL content, two additional iongel membranes were prepared with 80 wt% [C_2_mim][TFSI] IL, and 0.5 and 7.5 wt% MMT, and the obtained CO_2_ permeabilities are illustrated in Figure 8. These specific MMT content were chosen based on the results obtained for the iongels containing 60 wt% IL, where the 60 TFSI-39.5 PEGDA-0.5 MMT iongel showed the highest gas selectivity, while the 60 TFSI-32.5 PEGDA-7.5 MMT iongel achieved the highest CO_2_ permeability. Remarkably, both free-standing iongel membranes reinforced with MMT and containing 80 wt% IL were able to withstand the gas permeation experiments, which had not been possible without the incorporation of MMT [12]. The CO_2_ permeabilities of the iongel membranes containing 80 wt% [C_2_mim][TFSI] IL were considerably higher than those of the iongels with 60 wt% IL content. At 0.5 wt% MMT content, the CO_2_ permeability increased from 60.3 to 115.3 barrer, while for iongel membranes reinforced with 7.5 wt% MMT it increased up to 132.3 barrer. This behavior can be attributed to the higher IL content, resulting in a higher affinity of the iongel towards CO_2_, as expected.

The influence of increasing the IL content in the ideal selectivities of the iongel membranes reinforced with 0.5 and 7.5 wt% MMT can be found in Figure 9a,b, respectively. For the iongel with 0.5 wt% MMT, a decrease in the CO_2_/N_2_ selectivity was obtained when the IL content was increased to 80 wt%. For CO_2_/CH_4_ and CO_2_/H_2,_ the variations in selectivity were not as significant. Interestingly, the opposite behavior was observed at the highest MMT content. For all gas pairs studied, the ideal selectivity increased with increasing IL content, at 7.5 wt% MMT. This is most likely attributed to the higher IL content in the iongel, which helps prevent the formation of agglomerates. A more even dispersion of MMT particles in the iongel matrix along with the higher solubility provided by the IL will promote CO_2_ transport, thus improving the selectivity.

### 3.7. Comparison with Other Iongel Membranes

In order to provide an overview of how the performance of the iongel membranes prepared in this work are comparable with other iongel membranes reported in the literature, the experimental results were plotted in the 2008 Robeson upper bounds for CO_2_/N_2_ and CO_2_/CH_4_ separation, as well as on the CO_2_/H_2_ upper bound developed by Rowe and co-workers, in Figure 10a–c, respectively [45,46]. The results for both two- and three-component iongels reported in the literature are also included [12,18,22,23,25,47,48,49,50,51,52,53,54,55,56,57,58,59,60,61,62,63,64,65,66,67,68,69,70,71,72,73].

It is clear that for the CO_2_/N_2_ and CO_2_/CH_4_ separations, the iongels reinforced with MMT generally present lower CO_2_ permeabilities and selectivities compared to those of other iongel membranes reported in the literature. The fact that MMT does not present affinity towards CO_2_ highly contributes to the low permeabilities and selectivities observed. On the other hand, for the CO_2_/H_2_ separation, the obtained results are placed among the ones reported in the literature.

It should be mentioned that even though the separation performances of the iongels reinforced with MMT are not as good as those reported for other iongels, the main goal of this work was to improve the mechanical stability of iongels, so they could be tested in their self-standing form. Different materials to prepare iongels should be considered, as long as their mechanical stability is preserved or enhanced.

## 4. Conclusions

In an attempt to improve the mechanical properties, PEGDA/[C_2_mim][TFSI] iongels reinforced with different contents of MMT nanoclay (0.2–7.5 wt%) were prepared. The influence of increasing MMT content on the thermal and mechanical stabilities, as well as on the gas transport properties of the iongels was assessed. Dense, flexible, and free-standing iongel membranes were successfully obtained, through a solvent-free UV polymerization process. Agglomerates were visible in the iongels, especially at higher MMT contents, however, no major defects were detected in the iongels structures. FTIR spectroscopy confirmed the high extent of the photopolymerization process of the PEGDA network. The thermal stability of the iongels was influenced by the incorporation of MMT particles, since the *T_onset_* decreased at MMT contents >5 wt%. Both puncture strength and elongation improved in the iongels reinforced with up to 1 wt% MMT (up to 6 MPa/mm and 16.8%, respectively), further decreasing at higher contents, due to the observed MMT agglomerations. A significant increase in gas diffusivity with increasing MMT content resulted in the highest CO_2_ permeability being achieved for the iongel containing 7.5 wt% MMT (98 barrer). On the other hand, the highest gas selectivities were obtained for the iongel reinforced with 0.5 wt% MMT. Due to the mechanical reinforcement provided by the MMT particles, it was possible to increase the IL content up to 80 wt%, which allowed for better dispersion of the MMT particles, increasing the ideal selectivity for all gas pairs, compared to the iongels composed of 60 wt% IL.

It is clear that the incorporation of MMT particles into the iongel is a suitable strategy to improve the mechanical stability of these materials, as long as a good dispersion is assured. Nevertheless, further research regarding different nanoparticles to improve the mechanical stability of iongels containing high IL contents should be carried out.

## Figures and Tables

**Figure 1 membranes-11-00998-f001:**
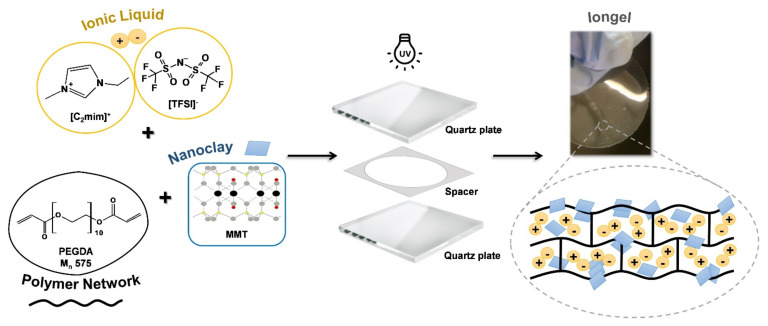
Schematic representation of the preparation of the iongels reinforced with MMT.

**Figure 2 membranes-11-00998-f002:**
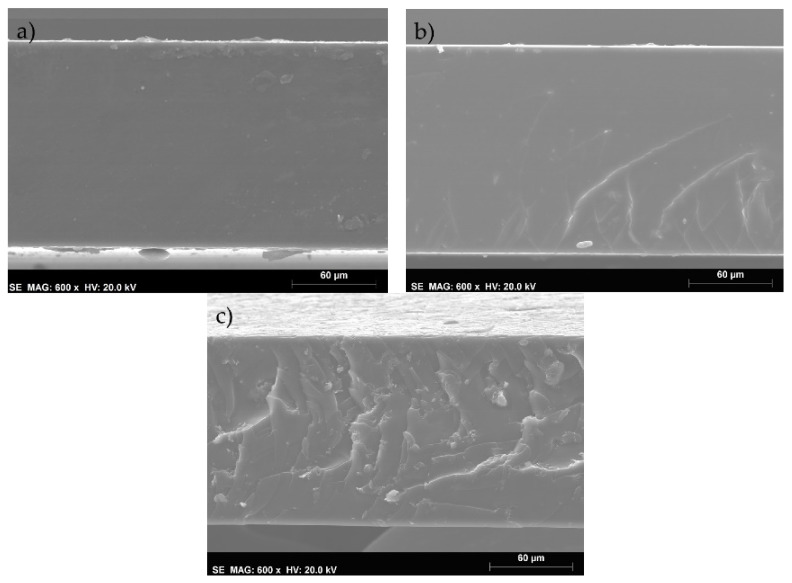
SEM images of the 60 TFSI-40 PEGDA iongel (**a**) and iongels reinforced with 0.2 (**b**) and 7.5 (**c**) wt% MMT.

**Figure 3 membranes-11-00998-f003:**
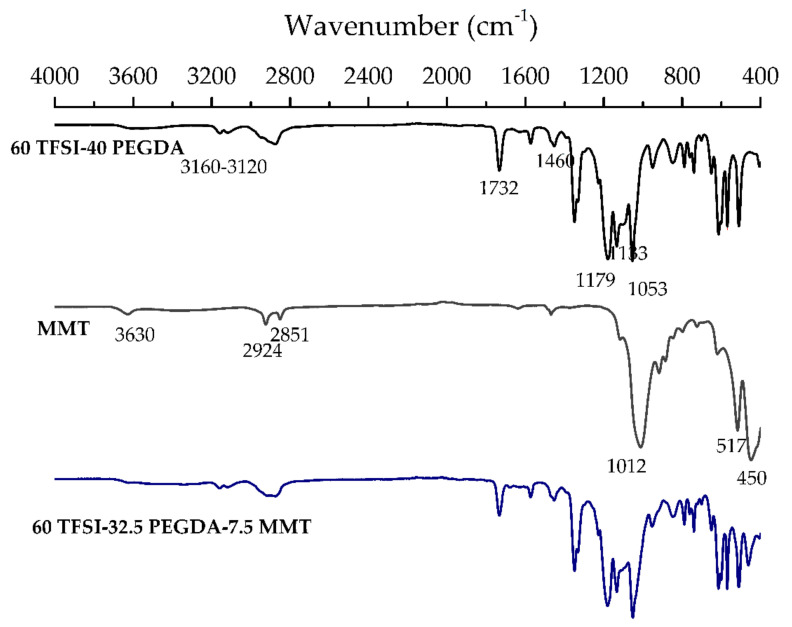
FTIR spectra of neat 60 TFSI-40 PEGDA iongel, MMT and 60 TFSI-32.5 PEGDA-7.5 MMT iongel.

**Figure 4 membranes-11-00998-f004:**
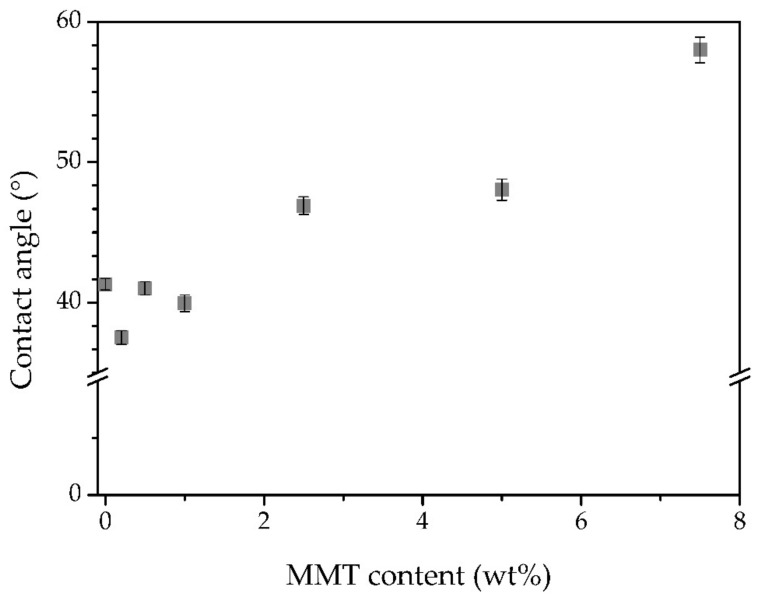
Water contact angles of all prepared iongels.

**Figure 5 membranes-11-00998-f005:**
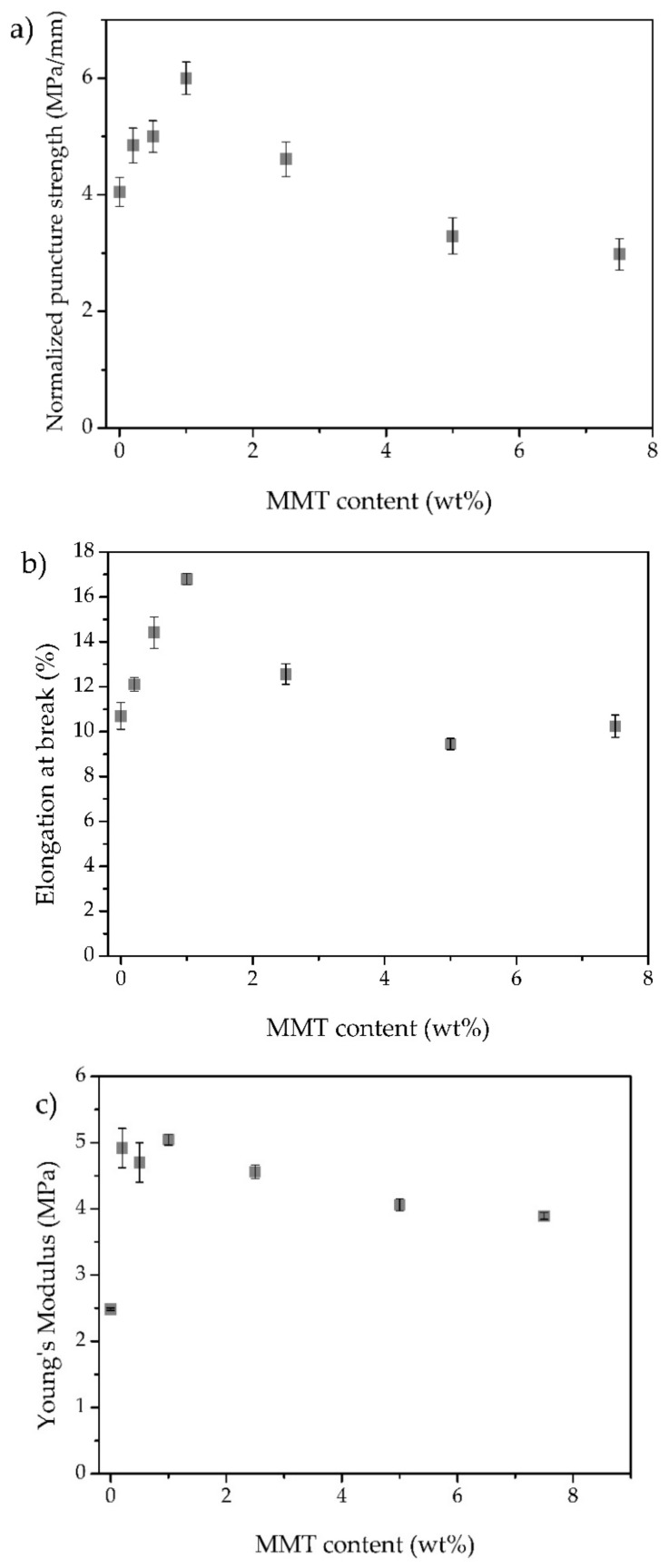
Normalized puncture strength (**a**), elongation at break (**b**) and Young’s Modulus (**c**) obtained for all prepared iongels, as a function of the MMT content.

**Figure 6 membranes-11-00998-f006:**
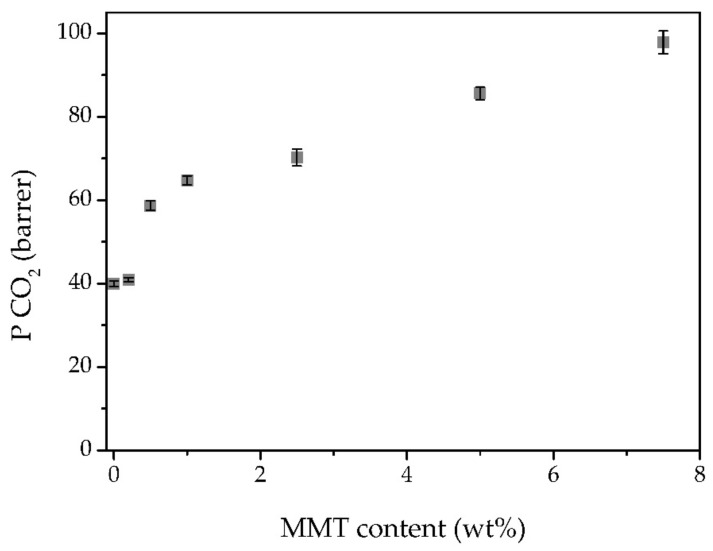
CO_2_ permeabilities obtained for all prepared iongels, as a function of the MMT content.

**Figure 7 membranes-11-00998-f007:**
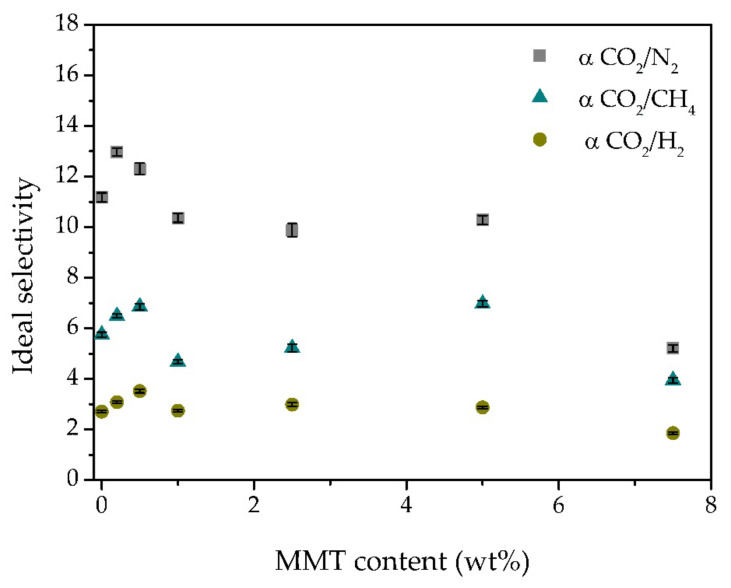
CO_2_/N_2_, CO_2_/CH_4_ and CO_2_/H_2_ ideal selectivities obtained for all prepared iongels, as a function of the MMT content.

**Figure 8 membranes-11-00998-f008:**
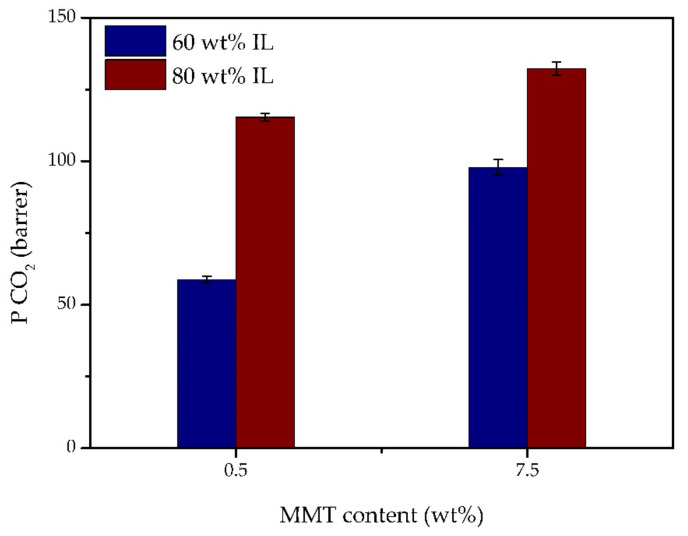
Influence of the IL content in the CO_2_ permeability of the iongels reinforced with 0.5 and 7.5 wt% MMT.

**Figure 9 membranes-11-00998-f009:**
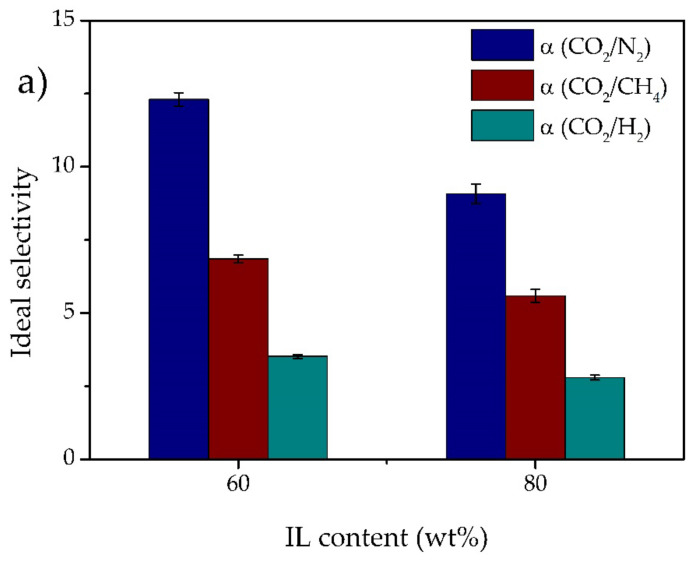
Influence of the IL content in the CO_2_/N_2_, CO_2_/CH_4_ and CO_2_/H_2_ ideal selectivities of the iongels reinforced with 0.5 (**a**) and 7.5 (**b**) wt% MMT.

**Figure 10 membranes-11-00998-f010:**
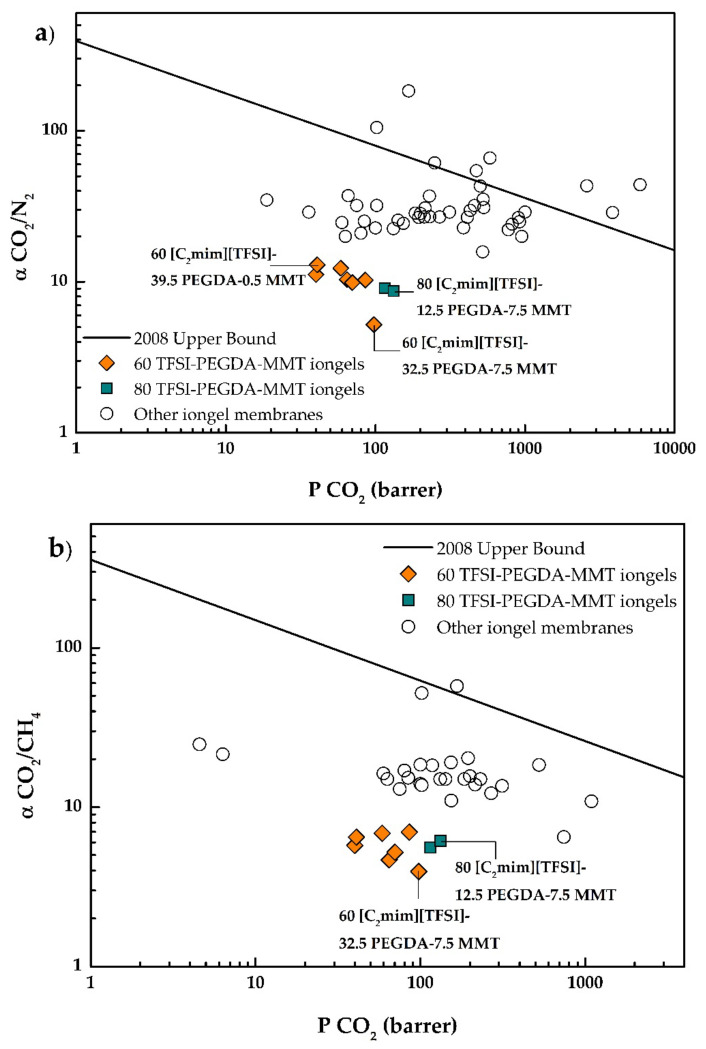
CO_2_/N_2_ (**a**), CO_2_/CH_4_ (**b**) and CO_2_/H_2_ (**c**) selectivities as a function of the CO_2_ permeability for all iongels prepared in this work, as well as other iongel membranes reported in the literature.

**Table 1 membranes-11-00998-t001:** Onset temperatures of the neat iongel components and the iongels reinforced with MMT.

Sample	T_onset_ (°C)
[C_2_mim][TFSI]	412
PEGDA	369
MMT	200
60 TFSI-40 PEGDA	339
60 TFSI-39.8 PEGDA-0.2 MMT	339
60 TFSI-39.5 PEGDA-0.5 MMT	340
60 TFSI-39 PEGDA-1 MMT	339
60 TFSI-37.5 PEGDA-2.5 MMT	340
60 TFSI-35 PEGDA-5 MMT	331
60 TFSI-32.5 PEGDA-7.5 MMT	326

## Data Availability

Not applicable.

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
