# Peer review of "Poly(ethylene glycol) Diacrylate Iongel Membranes Reinforced with Nanoclays for CO2 Separation"

_membranes, 2021, doi:10.3390/membranes11120998_

Round 1

Reviewer 1 Report

The manuscript is well written with clear explanations.

- In the introduction section, it's better to provide an overview of the materials used for other gas separation applications (other than carbon dioxide)

- What about long-term separation studies? Does the aging of membranes over time will not adversely affect selectivity? Do you have any experimental details on it?

- It's always better to compare with other membranes published in the literature at the end of the manuscript. This will provide the reader an overview of how well your material performs compared to other materials

- The SEM images in the main manuscript should be improved (at least play with contrast sharpness and brightness) and provide the visible scale bar.

Author Response

Dear Reviewer,

Please find in the attach file our reply to each comment and suggestion.

Kind regards,

Luísa Neves 

Reviewer 2 Report

Authors present a well written and interesting manuscript on gas separation processes with iongel membranes reinforced with nanoclays.

The work is of interest as the selected ionic liquid, also studied in many other research areas. Therefore, I recommend the publication of the present manuscript after the following issues have been addressed.

Was the IL purified at vacuum? Even APILs can present quantities of water that can vary its properties

Sharpness of Fig 1 could be improved

Line 153, t onset and t dec, what is the difference? Since Tdec is introduced on abbreviations and methodology, and it is not used along the manuscript.

Line 261 instead of Tonset, “defined as”, would be more appropriate “considered as”, since there are different definitions for tonset

Onset temperature should be compared to other published papers focussed on thermophysical characterization of pure ILs, that studied different experimental conditions, and the difference of 10 to 20 degrees with the authors could be explained based on experimental conditions or and the way of onset temperature determination. (http://dx.doi.org/10.1016/j.jct.2017.04.016, https://doi.org/10.1021/jp044626d)

Authors should try symbols on Figure S3, at least on the pure compounds figure since it is not clear enough.

Is it really needed Figure S5 of supplementary materials?

Would have been interesting a DSC study to determine transition temperatures at temperatures lower than 25 ºC?

Author Response

Dear reviewer,

Please find in attach reply to all your comments and suggestions.

Kind regards,

Luísa Neves 
